# Resveratrol Analogs and Prodrugs Differently Affect the Survival of Breast Cancer Cells Impairing Estrogen/Estrogen Receptor α/Neuroglobin Pathway

**DOI:** 10.3390/ijms24032148

**Published:** 2023-01-21

**Authors:** Emiliano Montalesi, Patrizio Cracco, Filippo Acconcia, Marco Fiocchetti, Giovanna Iucci, Chiara Battocchio, Elisabetta Orlandini, Lidia Ciccone, Susanna Nencetti, Maurizio Muzzi, Sandra Moreno, Iole Venditti, Maria Marino

**Affiliations:** 1Department of Science, University Roma Tre, V.le G. Marconi, 446, 00146 Rome, Italy; 2IRCCS Fondazione Santa Lucia, Via del Fosso di Fiorano, 64, 00179, Rome, Italy; 3Department of Earth Science, University of Pisa, Via Santa Maria, 53, 56126 Pisa, Italy; 4Research Centre E. Piaggio, University of Pisa, 56122 Pisa, Italy; 5Department of Pharmacy, University of Pisa, Via Bonanno, 6, 56126 Pisa, Italy

**Keywords:** breast cancer cells, estrogen receptor α, resveratrol, resveratrol analogs, gold nanoparticles

## Abstract

Breast cancer is the first leading tumor in women in terms of incidence worldwide. Seventy percent of cases are estrogen receptor (ER) α-positive. In these malignancies, 17β-estradiol (E2) via ERα increases the levels of neuroglobin (NGB), a compensatory protein that protects cancer cells from stress-induced apoptosis, including chemotherapeutic drug treatment. Our previous data indicate that resveratrol (RSV), a plant-derived polyphenol, prevents E2/ERα-induced NGB accumulation in this cellular context, making E2-dependent breast cancer cells more prone to apoptosis. Unfortunately, RSV is readily metabolized, thus preventing its effectiveness. Here, four different RSV analogs have been developed, and their effect on the ERα/NGB pathway has been compared with RSV conjugated with highly hydrophilic gold nanoparticles as prodrug to evaluate if RSV derivatives maintain the breast cancer cells’ susceptibility to the chemotherapeutic drug paclitaxel as the original compound. Results demonstrate that RSV conjugation with gold nanoparticles increases RSV efficacy, with respect to RSV analogues, reducing NGB levels and enhancing the pro-apoptotic action of paclitaxel, even preventing the anti-apoptotic action exerted by E2 treatment on these cells. Overall, RSV conjugation with gold nanoparticles makes this complex a promising agent for medical application in breast cancer treatment.

## 1. Introduction

Global cancer statistics estimated that female breast cancer in 2020 surpassed lung cancer as the most diagnosed malignancy, with 2.3 million new cases (11.7%) worldwide, followed by lung (11.4%), colorectal (10.0%), prostate (7.3%), and stomach (5.6%) cancers [1]. Although breast cancer is a highly heterogeneous disease, more than 70% of breast cancers are dependent on estrogen (E2) and are classified as estrogen receptor α (ERα)-positive. This subtype of breast cancer is more common in young than in post-menopausal women and results more prone to relapse and drug resistance than its ERα-negative counterparts [2,3,4]. We previously demonstrated that several E2-induced actions in ERα-positive breast cancers strongly rely on the activation of a specific E2/ERα/Neuroglobin (NGB) axis, leading to NGB accumulation and its mitochondrial location. Such redistribution prevents cytochrome c release into the cytosol, thus reducing susceptibility to apoptotic induction [5,6]. Intriguingly, E2 protection from the pro-apoptotic action of the chemotherapeutic drug paclitaxel in ERα-positive MCF-7 cells is strongly impaired by NGB knockdown [6]. These data raise the hypothesis that specific ERα antagonists preventing the increase in NGB levels may enhance the susceptibility of breast cancer cells to the action of chemotherapeutics.

Several studies have investigated the antioxidant, anticancer, and anti-inflammatory activities of resveratrol (RSV), a plant-derived polyphenol belonging to the group of stilbenoids [7]. Specifically, RSV binds to both ER subtypes exhibiting E2 antagonist activity for ERα [8]. Moreover, our previous studies have demonstrated that RSV prevents ERα signaling, downregulating NGB intracellular levels. This effect results in an effective strategy to increase MCF-7 breast cancer cells’ susceptibility to the chemotherapeutic drug paclitaxel and to impair E2 proliferative effects on breast cancer cells [9].

Despite its promising anti-cancer properties, RSV use in clinical practice is hindered by several limitations. In particular, in vivo effects of RSV appear to be affected by its low solubility and low bioavailability [7,10,11], due to its rapid metabolism, which leads to the co-presence of a plethora of metabolites that surpass concentrations of the original compound [12,13]. Effects of these metabolites might differ from those exerted by their precursor, potentially impairing its action, as demonstrated for daidzein [14]. Such a scenario is made even more complex if one considers that RSV effective dosage in vitro (µM range in cell culture media) is hardly reached in vivo (nM range in the blood), thus making it difficult to identify the adequate dosage of this compound to be administered to human subjects [15].

To date, the majority of efforts aim to improve the pharmacokinetic properties and/or enhance the bio-efficacy of RSV. Thus, according to the literature, a large number of RSV derivatives, grouped into (1) synthetic analogs and (2) prodrugs, have been developed and reported in the literature ([15] and literature cited therein). RSV analogs are derivatives obtained through structural modification on the trans-stilbene core that may enhance RSV biological and pharmacological activity. Differently, RSV prodrugs maintain the 3,5,4′-trihydroxystilbene skeleton in their structure and could modulate RSV absorption, distribution, metabolism, and excretion.

In the present study, four different RSV analogs were developed and their effects on ERα/NGB pathway were compared with RSV-conjugated gold nanospheres as a prodrug to evaluate if RSV derivatives maintain breast cancer cell susceptibility to the chemotherapeutic drug paclitaxel as the non-conjugated compound.

## 2. Results

### 2.1. Effect of Resveratrol Analogs on NGB Levels in MCF-7 Breast Cancer Cells

Four different RSV analogs, called RAV1, RAV2, RAV3, and NS012 (Figure 1) were synthesized as described in the materials and methods section. MFC-7 breast cancer cells were stimulated with three concentrations of the RSV analogs (i.e., 0.1, 1.0, and 10.0 μM) for 24 h, and the levels of NGB were evaluated (Figure 2). E2 (10 nM) was used as a positive control for the induction of NGB levels [9]. Since we wanted to evaluate the possible potentiation of these products in RSV chemical manipulation, the polyphenol was used at the lowest concentration tested in our previous studies (0.1 μM), which we observed to be unable to downregulate NGB [9]. As reported in Figure 2, 24 h of E2 treatment enhanced NGB levels, while, as expected, the low concentration of unconjugated RSV did not modify the protein levels. The RSV analogs showed a different effect in NGB expression. Indeed, RAV1 0.1 and 1.0 μM increased NGB levels, while RAV2 reduced NGB basal levels only at 0.1 μM concentration, and neither RAV3 or NS012 modified NGB levels even at high concentrations (i.e., 10 μM), at which unconjugated RSV reduced NGB levels [9].

### 2.2. Effect of Gold Nanoparticle-Conjugated RSV on NGB Levels Modulation

The results obtained with RSV synthetic analogs prompted us to investigate the effects of RSV conjugation with nanocarriers as prodrugs. Recently, RSV conjugated with gold nanoparticles has been efficiently synthesized and characterized in terms of toxicity in MCF-7 cells [16]. According to these previous results, the tested concentrations of RSV in conjugation with gold nanoparticles (NP-R) were 0.01, 0.03, and 0.1 μM, corresponding to a concentration of unconjugated nanoparticles (NP) of 1.0, 3.0, and 9.1 μg/mL.

The ability of NP-R and NP to modulate NGB levels has been tested on two ERα-positive breast cancer cell lines (i.e., MCF-7 and T47D), and one ERα-negative breast cancer cell line (i.e., MDA-MB-231) [17]. E2 10 nM has been used as the positive control. At 0.1 μM concentration, NP-R exerted an efficient downregulation of NGB levels in both ERα-positive MCF-7 and T47D cells (Figure 3A,B), while no modulatory effect on the protein was observed in ERα-negative MDA-MB-231 (Figure 3C). Unconjugated NP did not modify NGB levels at any used concentration in all tested cell lines (Figure 3).

### 2.3. Involvement of ERα and ERα Signaling in NP-R Effects

The lack of any effect of NP-R on NGB levels in ER negative cells suggests that, like RSV, ERα expression is pivotal for a NP-R-induced NGB decrease. To further confirm this evidence, MCF-7 cells have been pre-treated with endoxifen 1 µM, a selective ERα inhibitor [18], before NP-R stimulation. As shown in Figure 4A, endoxifen pre-treatment completely impairs both E2 and NP-R effects, strongly corroborating the involvement of ERα in action mechanism of gold-conjugated RSV. ERα, like other members of the nuclear receptor superfamily, is a ligand-dependent transcription factor that regulates the expression of genes containing the estrogen-responsive element (ERE) in their promotors (e.g., *cathepsin D*, *pS2*) [19]. However, upon ligand binding, ERα rapidly activates extranuclear signals important for the transcription of non-ERE containing genes (e.g., *cyclin D1*), which are important for E2 proliferative effects [19,20]. To evaluate the impact of NP-R on ERα activities, MCF-7 cells were stimulated for 24 h with the most efficient concentrations [9] of E2 (10 nM, positive control), unconjugated RSV (1.0 µM), and NP-R (0.1 µM) in the presence or absence of E2 and the levels of protein codified by non-ERE- (i.e., *cyclin D1*) and ERE-containing (i.e., *cathepsin D* and *pS2*) genes were determined. Figure 4B shows that, unlike E2, neither RSV or NP-R can increase cyclin D1 levels, but both molecules act as antagonists of E2, impairing the hormone effect on cyclin D1. Notably, NP-R treatment is more efficient than unconjugated RSV, reducing cyclin D1 level by 50% at 0.1 µM concentration. Surprisingly, when looking at cathepsin D and pS2 protein modulation, whose genes contain ERE box in the promoter and which are induced by E2, both unconjugated RSV and NP-R acted as E2 mimetics (Figure 4C,D).

### 2.4. Functional Outcomes of NP-R

We previously demonstrated that the ERα/NGB pathway is a compensatory mechanism triggered by E2 to increase the survival of breast cancer cells against different stressors, including chemotherapeutic drugs [6,17,21]. This evidence, together with the NP-R ability to impair NGB accumulation in cancer cells, prompted us to evaluate the capability of the prodrug to render breast cancer cells more prone to the anticancer effect of the chemotherapeutic drug paclitaxel.

Figure 5 shows that, in E2-responsive breast cancer cells (i.e., MCF-7 and T47D), pre-stimulation with NP-R (0.1 µM) potentiates paclitaxel action in reducing NGB levels. The treatment was also effective in preventing the E2-mediated inhibitory action exerted on NGB modulation (Figure 5A,B). This effect of NP-R is observed at a concentration that is 10 times lower than that required for the unconjugated polyphenol [9]. This potentiated action of NP-R on NGB levels was paralleled by the increase in the poly (ADP-ribose) polymerase-1 (PARP-1) cleavage (i.e., 89 kDa band), a known late apoptosis biomarker. Indeed, NP-R stimulation of MCF-7 cells increased the pro-apoptotic action of paclitaxel, also reducing the E2 protective effect against the chemotherapeutic agent (Figure 5C).

### 2.5. Cellular Uptake and Internalization of NP-R

To address the mechanisms underlying the greater efficiency of NP-R compared to unconjugated resveratrol, we investigated whether NP-Rs are directly internalized by the cell. To this purpose, cells treated with either unconjugated or conjugated resveratrol, and nanoparticles were ultrastructurally analyzed by Focused Ion Beam/Scanning Electron Microscopy (FIB/SEM). Treatment with a vehicle was used as a control. As shown in Figure 6D, a large number of nano-particles were localized to the cytosol after 24 h of treatment in samples treated with 0.1 µM NP or NP-R. Both naked and conjugated nanoparticles displayed the tendency to aggregate with some of the largest macrocomplexes located inside vacuoles for their possible elimination. It is noteworthy that cells exposed to nanoparticles exhibited similar ultrastructural features to control cells. In fact, in all the analyzed conditions, cells showed a regular plasma membrane with microvilli, a rounded nucleus with a smooth or slightly irregular nuclear envelope, and abundant mitochondria displaying an even distribution throughout the cytoplasm and regular cristae arrangement. The remaining organelles (e.g., endoplasmic reticulum and Golgi apparatus) showed normal ultrastructural features and cytoplasmic distribution irrespective of the different conditions investigated.

## 3. Discussion

In the last few years, a great deal of attention has been paid to the natural polyphenolic compounds due to their many biological effects [22,23,24] Among others, the stilbene trans-RSV, a polyphenolic compound which occurs abundantly in peanuts, red wine, and other vegetal sources, received particular attention. Several beneficial anticancer effects of RSV are reported in the literature, relying both on its antioxidant properties and other molecular mechanisms, such as DNA repair, cell cycle modulation, and estrogenic/antiestrogenic activities [25,26,27,28,29,30]. However, RSV effects have never been translated into any clinical trial, probably because the RSV concentrations required to obtain in vitro effects, ranging from 50 to 300 μM, are not reached in human plasma due to very low RSV bioavailability [31,32]. RSV biotransformation operated by enzymes of small intestine epithelial cells, liver, and even gut flora, not only reduces its concentration and persistence in the human body, but also leads to the co-presence in the bloodstream of numerous metabolites, which could have different effects on cancer cells, as reported for other polyphenols [14]. The activity of both enteral and hepatic enzymes is so extensive that only 1% of RSV is found in plasma as free substance. Based on current knowledge of the molecule bioavailability, an adequate dosage of RSV in human beings would correspond to about 150 mg/day, whereas currently, marketed food supplements contain about 20 times less RSV. A specific strategy is, therefore, needed to preserve RSV from the extensive metabolism to which polyphenols are exposed, and to maintain its anticancer effects [33,34]. A common way to increase the bioavailability of drugs is to improve the aqueous solubility of compounds through their functionalization [35]. Several kinds of RSV functionalization have been reported in the literature and a variety of chemically different capping groups and pro-moieties have been used to modulate the absorption and release of this molecule, as well as to improve its biological activity [15,34]. In this study, two different approaches were evaluated: the use of chemically modified RSV analogs and RSV conjugation with highly hydrophilic gold nanospheres as nanocarriers.

RSV analogs, RAV1, RAV2, RAV3, and NS012, were synthesized (see Section 4.2 of material and methods) to obtain new resveratrol analogs with improved bioavailability with respect to RSV. An etheroatomic linker endowed with major flexibility replaced the stilbene moiety. The 3,5 di hydroxyl phenyl group was substituted with the 1,3-benzodioxole moiety, while the para hydroxyl phenyl portion was either maintained as in NS012 or replaced with a 1,3-benzodioxole moiety (RAV2), with aromatic ring functionalized by iodine and hydroxyl groups (RAV3), or with a more lipophilic group (RAV1). These modifications impinge on RSV chemical structure, influencing the stilbene bioavailability [15,36]. However, despite the explored modifications, none of the analogs modulate NGB levels any more effectively than free RSV. RAV2 was the only molecule with a downregulating action on the protein; however, this effect was not dose-dependent and was observed only at 0.1 μM RAV2 concentration. The divergence in the action of RAV2 and RSV seems to indicate that this analog does not act on NGB through the same pathways as RSV. Moreover, the chemical structure of RSV analogs could allow them to be exposed to biotransformation enzymes.

RSV conjugation with highly hydrophilic gold nanospheres as nanocarriers was tested as another strategy to increase RSV efficacy and bioavailability. To ensure high hydrophilicity and biocompatibility in the transport system, citrate and L cysteine were used as ligands on the gold nanoparticles’ surface. Specifically, cysteine is the only amino acid to have a thiol group, an amino group, and a carboxyl group, which creates several possibilities for bonding to the metal surface [37,38,39]. Moreover, the organic−metal coupling may vary substantially and exhibit different configurations. Several studies proposed a bilayer cysteine boundary around metal nanoparticles: the first layer is made of cysteine molecules forming covalent S−metal (i.e., S-Au) bonds with the nanoparticle surface and has charged amino and carboxylate groups oriented outward. The outer layer interacts with the inner layer through hydrogen-bond intermolecular forces and has the thiol (S-H) groups pointing outward [40].

The toxicity of naked and RSV conjugated nanocarriers (NPs and NPs-R) was assessed in previous studies [16]. On this premise, a concentration of 9.1 µg/mL, corresponding to an RSV load of 0.1 μM, was indicated as safe and used to evaluate NGB modulatory action on breast cancer cells expressing different ERα levels. Free RSV does not reduce NGB content in MCF-7 cancer cells when in concentrations lower than 1μM, nor could it have that effect on T47D cells [9], whose ERα expression is smaller than other E2-dependent cancer cells [17]. On the other hand, when conjugated with gold nanospheres, RSV downregulates NGB levels in both cell lines, even at 0.1 μM concentration. ERα expression is strictly required for NP-R effect on NGB level modulation; indeed, this action is not observed in ERα-negative MDA-MB-231 cell lines. In addition, gold-conjugated RSV does not reduce NGB levels in co-stimulation with specific ERα antagonist endoxifen. Free RSV and NP-R activate the expression of estrogenic markers pS2 and cathepsin D proteins, whose gene transcription relies on the ERE signaling of ERα [41,42]. On the other hand, the same compounds act as Cyclin D1 protein stabilizers (not altering its basal levels, but preventing E2-induced Cyclin D1 increase), which, in turn, is modulated by rapid ERα signaling [20]. Overall, these data demonstrate that NP-Rs maintain the antagonistic effect of RSV on rapid ERα signaling without affecting ERα transcriptional activity. The action of polyphenols as partial ERα antagonists has been reported for the flavonoids naringenin and quercetin. Even in the presence of E2, both of these substances impair ERα-activated rapid signaling (i.e., receptor palmitoylation, ERK/MAPK, and PI3K/AKT activation), which is important for the cyclin D1 expression and, consequently, for cell cycle progression, without affecting ERα transcriptional activity [43]. Present data indicate that this mechanism is also activated by NP-R, which inhibits only rapid ERα pathways, committed to proliferation, as previously reported [16].

Intriguingly, cellular uptake analyses reveal that NP-R can be transported into breast cancer cells. The presence of gold aggregates distributed both in the cytosol and within vacuoles is consistent with the literature data [44,45], reporting that NPs internalization usually occurs via endocytosis/macropinocytosis mechanisms [46,47]. Besides confirming efficient NPs uptake, our obtained data, following a relatively long incubation time (24 h), suggest the involvement of diverse vesicular trafficking pathways, possibly including autophagocytic and/or exocytotic processes.

To assess the maintenance of the protective action of NP-R in increasing the pro-apoptotic action of the chemotherapeutic drug paclitaxel, MCF-7 and T47D were co-stimulated with NP-R and paclitaxel either in the presence or in absence of E2. In both cell lines, a 0.1 μM concentration of the compound potentiates paclitaxel-induced NGB downregulation, even in presence of E2. Moreover, the pro-apoptotic potentiation of the chemotherapeutic drug on MCF-7 cells was evaluated. NP-R enhances paclitaxel pro-apoptotic action and avoids E2-mediated protective effects at 0.1 μM, a concentration easily reached in the bloodstream. These data demonstrate that NP-R maintains both the mechanisms and the functional outcome of free RSV, as reported previously [9].

Overall, the use of highly hydrophilic gold nanocarriers improved RSV efficacy in reducing NGB levels and the resulting potentiation of the paclitaxel effect, probably reducing the undesired toxic effects of the chemotherapeutic drug. In addition, RSV’s improved activity did not lead to proliferative or tumor-promoting activity. Indeed, the physiological outcome of the augmented effectiveness of the compound was still related to cancer suppression. Although in vivo data are necessary and our laboratory is active on this issue, obtained data indicate that the developed formulation of NP-R not only could enhance RSV bioavailability but also strengthen its bioactivity, providing encouraging outcomes for the translation of these different administration routes into clinical application for breast cancer treatment.

## 4. Materials and Methods

### 4.1. Reagents

Sodium Citrate (Na_3_C_6_H_5_O_7_), L-Cysteine (L-cys), (C_3_H_7_NO_2_S), tetrachloroauric(III) acid trihydrate (HauCl_4_·3H_2_O), sodium borohydride (NaBH_4_), and Resveratrol (RSV) were used as received (Merck, Darmstad, D). The Bradford protein assay was obtained from Bio-Rad Laboratories (Hercules, CA, USA). The anti-α-tubulin and antivinculin were purchased from Sigma-Aldrich (St. Louis, MO, USA). Specific antibodies against poly[ADP-ribose] polymerase 1 (PARP-1), pS2, cathepsin D (CAT D), and Cyclin D1 were obtained from Santa Cruz Biotechnology (Santa Cruz, CA, USA). The anti-NGB antibody was purchased from Merck Millipore (Burlington, MA, USA). The ERα inhibitor endoxifen was purchased from Tocris (Ballwin, MO, USA). The chemotherapeutic drug paclitaxel (Pacl) was purchased from Sigma-Aldrich (St. Louis, MO, USA). The chemiluminescence reagent for Western blot ECL was obtained from GE Healthcare (Little Chalfont, UK). All the other products were obtained from Merck (Darmstad, D). Analytical and reagent grade products were used without further purification.

### 4.2. Synthesis of RSV Analogues

The synthesis of RSV analogues (RAV1, RAV2, RAV3, and NS012) was carried out as illustrated in Figure 1.

The *O*-(benzo[d][1,3]dioxol-5-ylmethyl)hydroxylamine hydrochloride was synthesized as previously described by a Mitsunobu reaction between the benzo[d][1,3]dioxol-5-ylmethanol, and the *N*-hydroxyphthalimide, followed by the removal of the protecting phthalimido group from the 2-(benzo[d][1,3]dioxol-5-ylmethoxy)isoindoline-1,3-dione, carried out with ammonia solution 7N in MeOH [48,49]. Treatment of hydroxylamine hydrochloride with the appropriate ketone or aldehydes, in methanolic solution and at room temperature (r.t.), furnished the opportune (*E*) *O*-benzyl oxime derivatives RAV1, RAV2, RAV3, and NS012 as the only geometric isomers. The correct configuration of RAV2, RAV3, and NS012 was determined by their ^1^HNMR spectra on the basis of the chemical shift value of the diagnostic iminic proton, ranging from 8.46 to 8.03 ppm, as reported by the literature for analogue compounds with *E* configuration [50,51].

Melting points (m.p.) were measured with a Leica Galen III microscope (Leica/Cambridge Instruments) and were uncorrected. ^1^H NMR spectra of compounds were recorded with a Bruker Ultrashield™ 400 MHz (Fällander, Switzerland) or Varian Gemini 200 MHz (Mountain View, CA, USA) spectrometer. Coupling constants *J* are reported in Hertz. The following abbreviations are used: singlet (s), doublet (d), doublet of doublet (dd), triplet (t), doublet of triplet (dt), broad (br), and multiplet (m). Reactions were monitored with thin layer chromatography (TLC) on silica gel plates containing a fluorescent indicator (Merck Silica Gel 60 F254), and spots were detected under UV light (254 nm). Chromatographic separations were performed on silica gel columns with flash column chromatography (Kieselgel 40, 0.040–0.063 mm; Merck). Na_2_SO_4_ was always used as the drying agent. Evaporation was carried out in vacuo (rotating evaporator). Elemental analyses were performed by our analytical laboratory and agreed with theoretical values to within ± 0.4%.

#### 4.2.1. General Procedure for the Synthesis of Derivatives RAV1, RAV2, RAV3, and NS012

To a solution of the commercially available ketone or aldehyde (0.382 mmol) in MeOH (4 mL), an aqueous solution (1 mL) of the O-(benzo[d][1,3]dioxol-5-ylmethyl)hydroxylamine hydrochloride (0.382 mmol) was added. The reaction mixture was stirred for 1 h at room temperature until the disappearance of the starting material (TLC analysis). The mixture was evaporated to dryness and the resulting crude was added to water and extracted with EtOAc. The organic phase was washed (3×) with water, dried (Na_2_SO_4_), filtered, and evaporated under reduced pressure to obtain a crude solid.

#### 4.2.2. (E)-1-(6-methoxynaphthalen-2-yl)ethan-1-one O-(benzo[d][1,3]dioxol-5-ylmethyl) Oxime (RAV1)

Compound RAV1 was obtained starting from the 1-(6-methoxynaphthalen-2-yl)ethan-1-one, following the general procedure. The crude was purified by crystallization (CHCl_3_/hexane) affording compound RAV1 as a white solid. Yield 50%. M.p.: 156–158 °C. ^1^H-NMR (400 MHz; CDCl_3_) δ: 7.91–7.85 (m, 2H, Ar); 7.74–7.67 (m, 2H, Ar); 7.14–7.11 (m, 2H, Ar); 6.95–6.78 (m, 3H, Ar); 5.95 (s, 2H, O-CH_2_-O); 5.15 (s, 2H, O-CH_2_); 3.93 (s, 3H, CH_3_); 2.35 (s, 3H, -OCH_3_). ^13^C-NMR (100 MHz; CDCl_3_) δ: 158.2, 154.8, 147.7, 147.0, 135.2, 131.9, 131.7, 129.9, 128.5, 126.8, 125.5, 124.0, 121.9, 119.0, 109.0, 108.09, 105.71, 100.97, 76.0, 55.3, 12.6. Anal. Calcd for C_21_H_19_NO_4_; C, 72.19; H, 5.48; N, 4.01; Found 72.30; H, 5.39; N, 4.39.

#### 4.2.3. (E)-benzo[d][1,3]dioxole-4-carbaldehyde O-(benzo[d][1,3]dioxol-5-ylmethyl) Oxime (RAV2)

Compound RAV2 was obtained starting from the benzo[d][1,3]dioxole-4-carbaldehydee, following the general procedure. The crude was purified by crystallization (CHCl_3_/hexane) affording compound RAV2 as a white solid. Yield 60%. M.p.: 142–144 °C. ^1^H-NMR (200 MHz; CDCl_3_) δ: 8.02 (s, 1H, HC=N); 7.21–7.20 (m, 1H, Ar); 6.99–6.87 (m, 5H, Ar) 5.99 (s, 2H, O-CH_2_-O); 5.96 (s, 2H, O-CH_2_-O); 5.07 (s, 2H, -OCH_2_). ^13^C-NMR (M100 Hz; CDCl_3_) δ: 153.0, 151.8, 148.7, 147.5, 134.2, 124.9, 121.5, 117.1, 116.2, 114.1, 112.8, 101.97, 76.9. Anal. Calcd for C_16_H_13_NO_5_; C, 64.21; H, 4.38; N, 4.68; Found 64.40; H, 4.08; N, 4.41.

#### 4.2.4. (E)-2-hydroxy-4-iodobenzaldehyde O-(benzo[d][1,3]dioxol-5-ylmethyl) Oxime (RAV3)

Compound RAV3 was obtained starting from the 2-hydroxy-4-iodobenzaldehyde, following the general procedure. The crude was purified by crystallization (CHCl_3_/hexane) affording compound RAV3 as a white solid. Yield 55%. M.p.: 136–137 °C. ^1^H-NMR (400 MHz; CDCl_3_) δ: 9.85 (s, 1H, OH); 8.08 (s, 1H, HC=N); 7.51–7.41 (m, 2H, Ar); 6.87–6.73 (m, 4H, Ar) 5.96 (s, 2H, O-CH_2_-O); 5.06 (s, 2H, -OCH_2_). ^13^C-NMR (M100 Hz; CDCl_3_) δ: 163.0, 157.0, 150.4, 149.0, 147.8, 139.6, 139.7, 129.9, 122.5, 119.1, 118.6, 109.1, 108.3, 101.1, 80.4. Anal. Calcd for C_15_H_12_INO_4_; C, 45.36; H, 3.05; N, 3.53; Found 45.66; H, 3.33; N, 3.40.

#### 4.2.5. (E)-4-hydroxybenzaldehyde O-(benzo[d][1,3]dioxol-5-ylmethyl) Oxime (NS012)

Compound NS012 was obtained starting from the 4-hydroxybenzaldehyde, following the general procedure. The crude was purified by crystallization (CHCl_3_/hexane) affording compound NS012 as a white solid. Yield 65%. M.p.: 123–125 °C. ^1^H-NMR (200 MHz; CDCl_3_) δ: 8.06 (s, 1H, HC=N); 7.50–7.46 (m, 2H, Ar); 6.93–6.78 (m, 5H, Ar) 5.96 (s, 2H, O-CH_2_-O); 5.08 (s, 2H, -OCH_2_). ^13^C-NMR (100 MHz; CDCl_3_) δ: 160.8, 155.3, 149.0, 135.6, 133.7, 124.9, 121.5, 116.1, 114.6, 112.1, 108.3, 101.1, 79.9. Anal. Calcd for C_15_H_13_NO_4_; C, 66.41; H, 4.83; N, 5.13; Found 66.01; H, 4.57; N, 5.02.

### 4.3. Synthesis and Purification of Gold NP and NP-R

The AuNPs stabilized with citrate and L-cys were prepared and characterized in analogy to literature reports [52]. Briefly: 25 mL of L-cys solution (0.002 M), 10 mL of citrate solution (0.01 M), and 2.5 mL of tetrachloroauric acid solutions (0.05 M) were mixed sequentially in a 100 mL flask, provided with a magnetic stir. After degassing with Argon for 10 min, 4 mL of sodium borohydride solution (0.00008 M) were added and the reaction continued for 2 h at room temperature. Then, the brown solid product was purified by centrifugation (13,000 rpm, 10 min, 4 times with deionized water).

NP-R synthesis was carried out following the same procedures, but including RSV water solution (1 mL 0.02 M) in the reagent mixture, before reduction.

### 4.4. Cell Culture

Human breast cancer cells MCF-7, T47D, and MDA-MB-231 were grown in air containing 5% CO_2_ in either modified, phenol red-free, or Dulbecco’s Modified Eagle’s Medium (DMEM) medium (MCF-7, T47D, MDA-MB-231). Ten percentage (*v*/*v*) of charcoal-stripped fetal calf serum, L-glutamine (2 mM), gentamicin (0.1 mg/mL), and penicillin (100 U/mL) were added to the media before use. Cells were used at passage 4–8, as previously described [5]. The cell line authentication was periodically performed by amplification of multiple short tandem repeat loci by BMR genomics S.r.l (Padova, Italy). Cells were treated for 24 h with either vehicle (ethanol [EtOH]/phosphate-buffered saline [PBS], 1:10; *v*/*v*) or E2 (1 or 10 nM) or RSV (1 µM) or Pacl (1, 10, and 100 nM) or NP (1.0, 3.0, 9.1 µg/mL) or NP-R (RSV concentration = 0.01, 0.03 and 0.1 μM). When indicated, endoxifen (1 µM) was added 30 min before NP-R administration, NP-R was added 1 h before E2 administration, or E2 (10 nM) was added 4 h before Pacl addition (0.1, 1, and 100 nM) for 24 h. The E2 concentrations were selected on the bases of dose–response experiments already performed [5,53].

### 4.5. Western Blot

Briefly, after the treatments, cells were lysed and proteins were solubilized in the YY buffer (50 mM HEPES at pH 7.5, 10% glycerol, 150 mM NaCl, 1% Triton X-100, 1 mM EDTA, and 1 mM EGTA) containing 0.70% (*w*/*v*) sodium dodecyl sulfate (SDS). Total proteins were quantified using the Bradford protein assay. Solubilized proteins (20 µg) were resolved by 7% or 15% SDS-polyacrylamide gel electrophoresis at 100 V for 1 h at 24.0 °C and then transferred to nitrocellulose with the Trans-Blot Turbo Transfer System (Bio-Rad) for 7 min. The nitrocellulose was treated with 5% (*w*/*v*) bovine serum albumin in 138.0 mM NaCl, 25.0 mM Tris, pH 8.0, at 24.0 °C for 1 h. Nitrocellulose was probed overnight at 4.0 °C with either anti-NGB (final dilution, 1:1000) or anti-PARP-1 (final dilution, 1:1000) or anti-pS2 (final dilution, 1:1000) or anti-catepsin D (final dilution, 1:1000) or anti-Cyclin D1 (final dilution, 1:1000) antibodies. The antibody reactivity was detected with ECL chemiluminescence Western blotting detection reagent using a ChemiDoc XRS + Imaging System (Bio-Rad Laboratories, Hercules, CA, USA). Densitometric analyses were performed by the ImageJ software for Microsoft Windows (National Institute of Health, Bethesda, Rockville, MD, USA).

### 4.6. Electron Microscopy

MCF-7 cells were grown on glass coverslips and fixed with 2% formaldehyde (from paraformaldehyde) and 1.25% glutaraldehyde in 0.1 M cacodylate buffer, pH 7.4, at 4 °C. The subsequent steps were performed on ice. After extensive washing in the same buffer, samples were post-fixed in 1% osmium, rinsed in distilled water, and stained with UranyLess (Electron Microscopy Science, Foster City, CA, USA) as a contrasting agent for 1 h in the dark. Samples were washed in distilled water, then gradually dehydrated in ethanol, to 100% ethanol. This step was repeated twice at room temperature, as were all subsequent steps. Cells were infiltrated with a 1:1 mixture of anhydrous ethanol and epoxy embedding medium (Sigma-Aldrich™, Cat# 45359-1EA-F), then in pure resin for 90 min. Excess resin was gently removed, prior to polymerization at 60 °C for 72 h. This delicate procedure allowed to readily identify the cell boundaries at FIB/SEM, facilitating the milling process and the sectioning of the sample. Resin-embedded coverslips were secured to stubs using an adhesive carbon disc and made conductive with a thin layer of gold with a K550 sputter coater (Emithech, Kent, UK). Resulting samples were analyzed with a Dual Beam (FIB/SEM) Helios Nanolab 600 (FEI Company, Hillsboro, OR, USA) at the electron microscopy interdepartmental facility (LIME) at Roma Tre University. The cells were cut by the ion beam operated at a voltage of 30 KV and a current of 6.5 nA. The resulting cross-sections were examined by the SEM column, detecting backscattered electrons, using a 2 KV voltage and a current of 0.17 nA.

### 4.7. Statistical Analysis

The statistical analysis was performed by the Student’s t test to compare two sets of data by INSTAT software system for Windows. In all cases, *p* < 0.05 was considered significant.

## Data Availability

The data presented in this study are available within the article.

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
