# Peer review of "Resveratrol Analogs and Prodrugs Differently Affect the Survival of Breast Cancer Cells Impairing Estrogen/Estrogen Receptor α/Neuroglobin Pathway"

_ijms, 2023, doi:10.3390/ijms24032148_

Round 1

Reviewer 1 Report

The manuscript is very interesting, well organized and has a rationale.

Perhaps a graphical abstract would be helpful to make the results immediate.

I suggest adding these manuscripts (see below) in the introduction to emphasize the importance of medicinal plants and their potential help for people's health.

Mitra S, Das R, Emran TB, Labib RK, Noor-E-Tabassum, Islam F, Sharma R, Ahmad
I, Nainu F, Chidambaram K, Alhumaydhi FA, Chandran D, Capasso R, Wilairatana P.
Diallyl Disulfide: A Bioactive Garlic Compound with Anticancer Potential. Front
Pharmacol. 2022 Aug 22;13:943967.

Küpeli Akkol E, Genç Y, Karpuz B, Sobarzo-Sánchez E, Capasso R. Coumarins and
Coumarin-Related Compounds in Pharmacotherapy of Cancer. Cancers (Basel). 2020
Jul 19;12(7):1959.

On what basis did the authors select the concentrations of resveratrol?

Do the authors think that the microbiota can interfere?

In the Discussion, the Authors should highlight the possible clinical significance of their findings

Author Response

The manuscript is very interesting, well organized and has a rationale.
Perhaps a graphical abstract would be helpful to make the results immediate.

Reply: We wish to thank the referee for his/her kind comment. We agree with the referee, but we had little time to submit the revisions, whereas the construction of a good graphical abstract requires a lot of time to avoid inaccuracy. Therefore, we decided not to include a graphical abstract.

I suggest adding these manuscripts (see below) in the introduction to emphasize the importance of medicinal plants and their potential help for people's health.
Mitra S, Das R, Emran TB, Labib RK, Noor-E-Tabassum, Islam F, Sharma R, Ahmad
I, Nainu F, Chidambaram K, Alhumaydhi FA, Chandran D, Capasso R, Wilairatana P.
Diallyl Disulfide: A Bioactive Garlic Compound with Anticancer Potential. Front
Pharmacol. 2022 Aug 22;13:943967.
Küpeli Akkol E, Genç Y, Karpuz B, Sobarzo-Sánchez E, Capasso R. Coumarins and
Coumarin-Related Compounds in Pharmacotherapy of Cancer. Cancers (Basel). 2020
Jul 19;12(7):1959.
Reply: These references have been added as references 22 and 23 in the discussion (page 7 line 217).

On what basis did the authors select the concentrations of resveratrol?

Reply: The resveratrol concentrations have been selected, basing on both literature (see references 8, 10, 11 and 15) and  our previous dose-response data (see reference 9), whereas the concentrations of resveratrol-conjugated with nanoparticles have been selected based on dose response experiments reported in figure 3.

Do the authors think that the microbiota can interfere?

Reply: Although it is difficult to predict what may happen in vivo, in our opinion the large steric hindrance imposed to resveratrol by the presence of nanoparticles makes this compound difficult to access the active site of the microbiota or intestinal and hepatic enzymes.

In the Discussion, the Authors should highlight the possible clinical significance of their findings.

Reply: Following the referee’s suggestion, a sentence on the possible clinical significance of our results has been added in the discussion.

In addition, the entire manuscript has been revised by an English mother tongue colleague.

Reviewer 2 Report

In this manuscript the authors tested the effects of four different RSV analogs that they developed and one RSV prodrug (RSV conjugation with gold nanoparticles) on the ERα/NGB pathway aiming to maintain the breast cancer cells susceptibility to the chemotherapeutic drug paclitaxel. The results of the RSV conjugated with gold nanoparticles are so promising as the hydrophilic gold nanocarriers improved RSV efficacy in reducing NGB levels and the resulting potentiation of paclitaxel effect, probably reducing the undesired toxic effects of the chemotherapeutic drug.

The study is so interesting, and the manuscript is clear, concise, and well-written. The introduction is relevant and gives sufficient information about the previous studies findings which leads to the current study rationale. The methods are generally appropriate and well presented. The results are clear and presented in a logic way. Overall, this is a high-quality manuscript. However, very minor comments are listed below:

1.     Line 43-44 Please review.

2.     Line 87 “NGB has been valuated” ïƒ  evaluated

3.     Line 157 C and D are different in the picture than in the caption.

4.     The resolution of Figure 5 needs to be adjusted.

Author Response

In this manuscript the authors tested the effects of four different RSV analogs that they developed and one RSV prodrug (RSV conjugation with gold nanoparticles) on the ERα/NGB pathway aiming to maintain the breast cancer cells susceptibility to the chemotherapeutic drug paclitaxel. The results of the RSV conjugated with gold nanoparticles are so promising as the hydrophilic gold nanocarriers improved RSV efficacy in reducing NGB levels and the resulting potentiation of paclitaxel effect, probably reducing the undesired toxic effects of the chemotherapeutic drug.

The study is so interesting, and the manuscript is clear, concise, and well-written. The introduction is relevant and gives sufficient information about the previous studies findings which leads to the current study rationale. The methods are generally appropriate and well presented. The results are clear and presented in a logic way.

Reply: The Authors thank the referee for its/her kind comments.

Overall, this is a high-quality manuscript. However, very minor comments are listed below:

  1. Line 43-44 Please review.
  2. Line 87 “NGB has been valuated”  evaluated
  3. Line 157 C and D are different in the picture than in the caption.
  4. The resolution of Figure 5 needs to be adjusted.

Reply: We wish to thank the referee for its/her suggestions that have been followed. In addition, the manuscript has been thoroughly revised by an English mother tongue colleague.